# A viral protease relocalizes in the presence of the vector to promote vector performance

Aurélie Bak[1], Andrea L. Cheung[1], Chunling Yang[2], Steven A. Whitham[2] & Clare L. Casteel[1]

Vector-borne pathogens influence host characteristics relevant to host–vector contact, increasing pathogen transmission and survival. Previously, we demonstrated that infection with *Turnip mosaic virus*, a member of one of the largest families of plant-infecting viruses, increases vector attraction and reproduction on infected hosts. These changes were due to a single viral protein, NIa-Pro. Here we show that NIa-Pro responds to the presence of the aphid vector during infection by relocalizing to the vacuole. Remarkably, vacuolar localization is required for NIa-Pro's ability to enhance aphid reproduction on host plants, vacuole localization disappears when aphids are removed, and this phenomenon occurs for another potyvirus, *Potato virus Y*, suggesting a conserved role for the protein in vector–host interactions. Taken together, these results suggest that potyviruses dynamically respond to the presence of their vectors, promoting insect performance and transmission only when needed.

[1] Department of Plant Pathology, University of California, Davis, California 95616, USA. [2] Department of Plant Pathology and Microbiology, Iowa State University, Ames, Iowa 50011, USA. Correspondence and requests for materials should be addressed to C.L.C. (email: ccasteel@ucdavis.edu).

Viruses dominate our planet, parasitizing all forms of life including other viruses. Because viruses are obligate parasites, they must encounter and then infect susceptible hosts to survive. After successful infection, transmission of a virus to the next host is a critical step for survival. Due to their limited mobility, many viruses, including ~80% of all plant-infecting viruses, rely on other organisms for transmission[1]. The specific organism that transmits a virus is known as a vector. Vectors are found among many different taxa, but for plant-infecting viruses they most often are hemipteran arthropods, and in particular aphids[2–4].

Because of their agricultural importance, plant viruses and their transmission have been the subject of research for more than a century. It has been documented that the yellow leaf colour and specific volatiles emitted by diseased plants modulate attractiveness for vectors (for review, see refs 5,6). In parallel, several studies have shown that viruses can suppress plant defences and thus vectors feeding on virus-infected plants often possess greater fitness than those residing on healthy plants (for review, see refs 5,7–10). These findings suggest that viruses may facilitate their own transmission by modulating host characteristics. Recently, it has been demonstrated that *Cauliflower mosaic virus* (CaMV) responds actively to the presence of its vector, changing transmission body morphs and increasing the potential for transmission[11]. This new concept in virology has been called 'perceptive viral behaviour' and relies on the fact that viruses respond directly or via the host to potential vectors[11].

*Turnip mosaic virus* (TuMV) is considered to be one of the most damaging viruses of vegetable crops worldwide[12–14], infecting >300 species of dicotyledonous plants[15]. TuMV is a member of the Potyviridae family and is non-persistently transmitted by >80 aphid species including *Myzus persicae* (green peach aphid)[16]. Its genome is a single ~10 kb RNA molecule that codes for a large polyprotein plus the pipo frameshift protein. The polyprotein is cleaved into 1 coat protein and 10 non-structural products via 3 virus-encoded proteinases named P1 protein (P1), Helper component proteinase (HC-Pro) and Nuclear inclusion protein a protease (NIa-Pro)[17–19]. During infection, NIa-Pro exists as a fully processed form and as a fusion with another TuMV protein, the viral genome-linked protein (VPg)[19–21]. The fusion contains VPg at the N terminus and NIa-Pro at the C terminus and is known as 'NIa' (Fig. 1a). Previously, we demonstrated that TuMV infection suppresses callose deposition, an important plant defence induced in response to aphid feeding, and increases aphid fecundity on infected plants. We determined that expression of NIa-Pro was responsible for decreased plant defences and increased aphid reproduction through changes in the phytohormone ethylene[9,22]. However, the cellular mechanisms by which NIa-Pro mediates changes in plant physiology remain unclear.

In this study, we investigate the subcellular localization of NIa-Pro and the function of localization in virus–vector–plant interactions. We demonstrate that NIa-Pro localizes in the cytoplasm and in the nucleus, similar to the precursor NIa (VPg:NIa-Pro) during infection[20]. However, in the presence of the insect vector, NIa-Pro relocalizes to the vacuole of the cell and this relocalization is essential for its ability to decrease plant defences and increase aphid performance on host plants. Surprisingly, when aphids are removed NIa-Pro disappears from the vacuole, suggesting that TuMV promotes vector performance only when needed. Furthermore, we demonstrate that this may be a more general phenomenon for potyviruses, one of the largest genera of plant-infecting RNA viruses[23]. These results demonstrate that NIa-Pro responds actively to the presence of the insect vector and links a biochemical and ecological function to the relocalization, a phenomenon never before demonstrated.

## Results

**The NIa-Pro protein is required to increase aphid fecundity.** Previously we demonstrated that TuMV NIa-Pro promotes aphid performance and suppresses plant defences[9]. To confirm that the NIa-Pro protein versus its RNA sequence mediates increased aphid fecundity on host plants, we made a new construct that contains the full sequence of the NIa-Pro open reading frame but prevents production of the functional protein in a green fluorescent protein (GFP)-expressing vector. The construct introduced one extra thymine immediately after the first codon of the NIa-Pro open reading frame, resulting in a frameshift mutation that generates a stop codon directly after the first codon, thus preventing production of the functional protein (GFP:NIa-Pro:Fs; Supplementary Fig. 1a–c). To determine whether NIa-Pro protein production is critical for enhancing aphid performance, we performed fecundity tests on *Nicotiana benthamiana* transiently expressing GFP:NIa-Pro, GFP:NIa-Pro:Fs or the empty expression vector (free GFP). Consistent with our previous findings, expression of NIa-Pro increased aphid

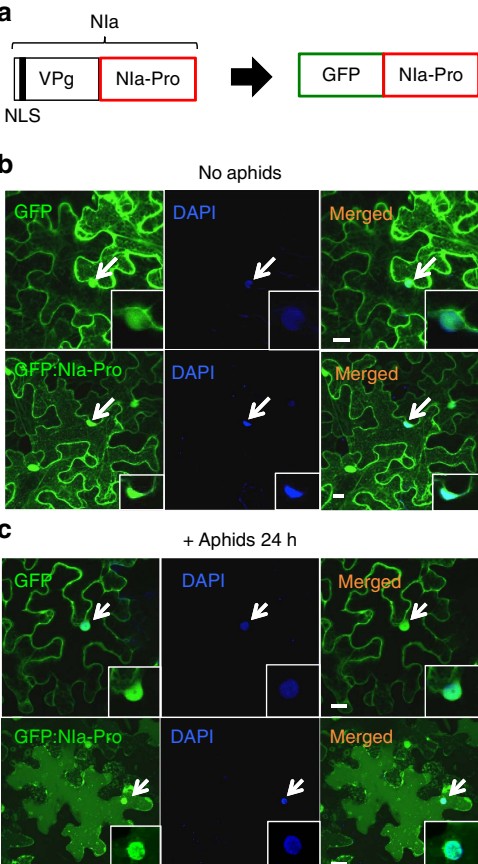

**Figure 1 | NIa-Pro relocalizes in the cell with aphids present.**
(**a**) NIa-Pro can exist as a complex with another TuMV protein, the viral genome-linked protein (VPg) or in isolation during infection. In our study, only NIa-Pro was used. Images show confocal projections of agroinfiltrated *N. benthamiana* leaves with free GFP and GFP:NIa-Pro (**b**) without and (**c**) with aphids present. The first panel on the left shows GFP and GFP:NIa-Pro fluorescence in green. The second panel in the middle shows the nucleus of the cell in blue that was counterstained with DAPI. The third panel represents a merge of the first two panels, demonstrating GFP and GFP:NIa-Pro localizes to the nucleus. The small insert panels represent single confocal sections. With the presence of aphids, GFP:NIa-Pro relocalizes in the cell, whereas free GFP does not. The nuclei are indicated with white arrows. Scale bars, 20 μm.

fecundity compared with aphids feeding on plants expressing the free GFP[9]. Inhibiting production of the NIa-Pro protein in the NIa-Pro:Fs mutant led to no increase in aphid fecundity (Supplementary Fig. 1d). This result demonstrates that host plant changes mediating increased aphid fecundity are conferred by the NIa-Pro protein and not by its RNA sequence.

**NIa-Pro relocalizes in the cell with aphids present**. To analyse the intracellular localization of NIa-Pro, we transiently expressed NIa-Pro fused to GFP under the control of the 35S promoter (GFP:NIa-Pro) in *N. benthamiana* plants (Fig. 1a). As a control we used the same construct without NIa-Pro in transient expression experiments (free GFP). GFP:NIa-Pro localized to the nucleus and to the cytoplasm of the cell, similar to the free GFP (Fig. 1b). Nuclear localization was verified using 4,6-diamidino-2-phenylindole (DAPI) staining[24] and by examining single sections on the confocal microscope (Fig. 1b, single sections shown in small panels). We next wanted to see if the presence of aphids on leaves alters NIa-Pro cellular localization because TuMV and NIa-Pro often occur with insect vectors in nature. To assess this, we allowed aphids to feed for 24 h on leaves transiently expressing free GFP or GFP:NIa-Pro. Surprisingly, aphids induced a relocalization of GFP:NIa-Pro in the cell, whereas free GFP localization did not change in the presence of the aphids (Fig. 1c).

**NIa-Pro relocalizes to the vacuole with aphids present**. GFP:NIa-Pro appeared to relocalize to the lumen of the vacuole in the presence of aphids[25]. To confirm vacuolar localization, we stained the leaves with SNARF-1, a marker for the vacuolar lumen[26]. As expected, when aphids were present GFP:NIa-Pro co-localized with the vacuole, whereas the free GFP control did not (see single sections; Fig. 2a,b). To further confirm localization to the lumen of the vacuole, we used a second marker, the spRFP:AFVY fusion protein[27]. Using this construct, GFP:NIa-Pro also colocalized with the vacuole when aphids were present, whereas the free GFP did not (see single sections; Fig. 2c,d). We next wanted to confirm that the phenotypes observed were not due to autofluorescence of dead cells using time-lapse imaging. We captured active trafficking of the free GFP control (cytoplasmic streaming) and the GFP:NIa-Pro (intra-vacuolar trafficking) verifying the cells were still alive (Supplementary Fig. 2). Previous studies have demonstrated that the GFP protein can be visualized in the plant vacuole[28,29]. Nevertheless, in a last control, we excited the GFP:NIa-Pro at 488 nm and performed a scan across 380–800 nm. We observed that the phenotypes observed plus or minus aphids were characteristic of the GFP emission (Supplementary Fig. 3). The scan revealed some chloroplast autofluorescence around 650 nm of emission.

**Vacuolar localization disappears when aphids are removed**. To determine the kinetics of NIa-Pro relocalization, we next allowed aphids to feed for different durations of time (2, 6 and 24 h) on leaves transiently expressing free GFP or GFP:NIa-Pro. After 2 h of aphid feeding, no relocalization was observed in either treatment, suggesting that insect probing may not be enough to induce the response (Fig. 3a, second panel). However, after 6 h of aphid feeding, cytoplasmic vesicles were observed in plants expressing GFP:NIa-Pro, while there was no change in phenotype for free GFP (Fig. 3a, third panel). After 24 h of aphid feeding, ~60% of the visible epidermal cells exhibited relocalization of NIa-Pro throughout the vacuole lumen (Fig. 3a, fourth panel; Fig. 3b). The large numbers of cells exhibiting the phenotype after 24 h of feeding suggest that the response is not restricted to cells probed by the aphids, but is instead mediated by a systemic signal. In

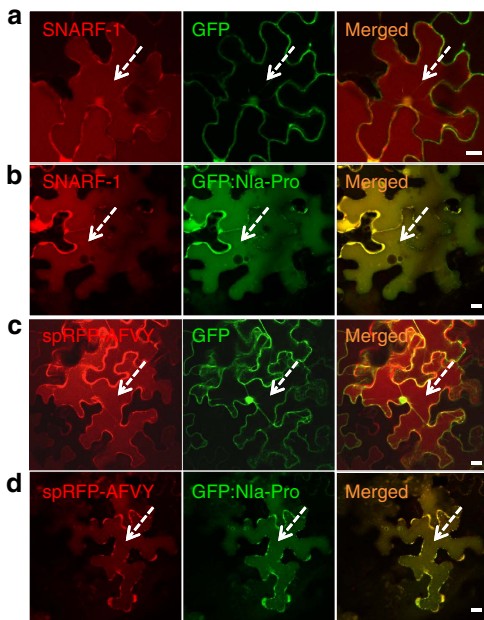

**Figure 2 | NIa-Pro relocalizes to the vacuole with aphids present.** Images show single sections of *N. benthamiana* leaves agroinfiltrated with (**a,c**) free GFP and (**b,d**) GFP:NIa-Pro with aphids present. (**b**) GFP:NIa-Pro co-localizes with the vacuole stained with the SNARF-1 dye in red (dashed arrows) in the presence of aphids, while (**a**) free GFP does not. (**c**) Free GFP and (**d**) GFP:NIa-Pro were coinfiltrated with spRFP:AFVY, a vacuolar luminal protein. (**d**) GFP:NIa-Pro co-localized with the lumen of the vacuole (dashed arrows) in the presence of aphids, while (**c**) free GFP did not. Scale bars, 20 μm.

parallel experiments, aphids were allowed to feed for 24 h and then removed. Leaves were allowed to recover for 2 h before microscopic imaging. Surprisingly, vacuolar localization of GFP:NIa-Pro was no longer observed 2 h after aphid removal (Fig. 3a, right panel), suggesting that GFP:NIa-Pro was able to relocalize to the nucleus or was degraded in the vacuole after aphids were removed. Quantification of three independent experiments established that aphid presence significantly modifies localization of GFP:NIa-Pro and nuclear localization is restored when aphids are removed (Fig. 3b).

Because free GFP is small enough to freely diffuse through the nuclear pore[30], we could not discount the fact that the GFP observed in the nucleus in the GFP:NIa-Pro experiments could be GFP cleaved from NIa-Pro. To verify that the GFP from the GFP:NIa-Pro fusion is not cleaved, we performed a western blot with a GFP antibody using the agroinfiltrated *N. benthamiana* leaves expressing free GFP, GFP:NIa-Pro with and without aphid feeding and after reversion (Fig. 3c). We found that GFP:NIa-Pro is not cleaved (Fig. 3c). Because GFP:NIa-Pro is ~54 kD and too large to diffuse into the nucleus, our data suggest that the nuclear accumulation of GFP:NIa-Pro is due to a property of NIa-Pro and not just because of its small size, like the free GFP[30].

**Relocalization is required to increase aphid fecundity**. To determine the functional significance of NIa-Pro relocalization in the presence of aphid vectors, we created two NIa-Pro localization mutants. For one mutant, we confined NIa-Pro to the nucleus by adding a strong nuclear localization signal (NLS; PKKKRKV) and for the other, we inhibited nuclear localization by adding a nuclear export signal (NES; NELALK-LAGLDINK) to the C-terminal end of NIa-Pro[31,32]. Localization

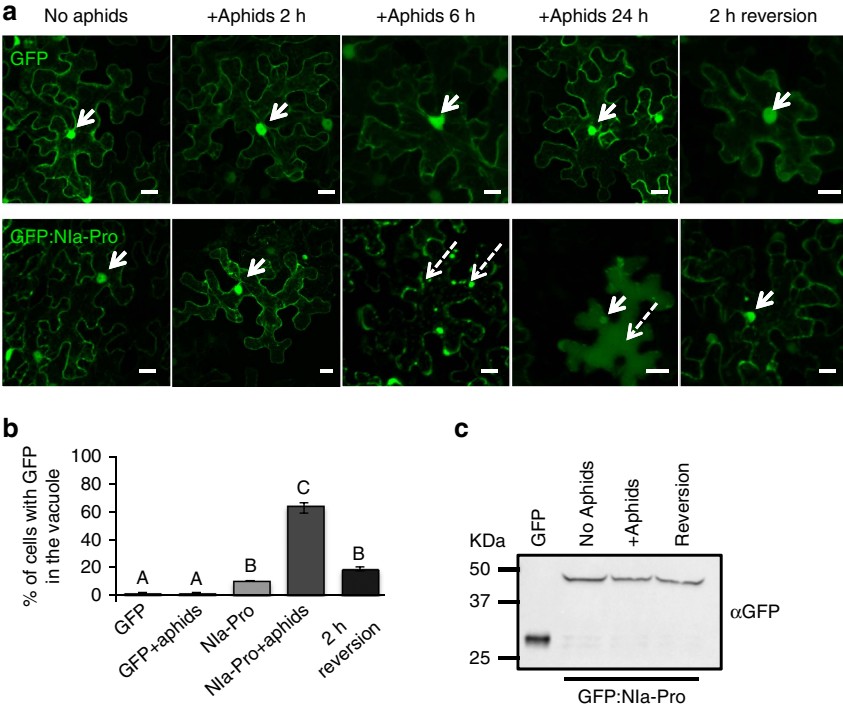

**Figure 3 | NIa-Pro localization to the vacuole occurs only when aphids are given access to plants.** (**a**) Images show confocal projections of
*N. benthamiana* agroinfiltrated leaves with free GFP (top panel) or GFP:NIa-Pro (lower panel). Separate leaves were infested with aphids for 2, 6 or 24 h
(middle panels) and a set of leaves were left uninfested as a control (left panel). For the reversion (right panel), all aphids were removed after 24 h of
infestation, and the leaves were observed after a 2 h recovery period. GFP:NIa-Pro relocalizes from the nucleus (white arrows) to form cytoplasmic
aggregates after 6 h (dashed arrows) and then diffuses into the vacuole after 24 h (dashed arrow). When the aphids were removed, GFP:NIa-Pro was
observed in the nucleus again (white arrow). The free GFP control does not show relocalization with any treatments. (**b**) The percentage of cells with GFP in
the cytoplasm for each of the treatments. (**c**) The image represents a western blot from leaf extracts expressing free GFP or GFP:NIa-Pro before aphid
infestation, after 24 h of aphid infestation or after the 2 h recovery period (reversion), and incubated with a GFP antibody. The western blot shows that the
phenotype in the presence of aphids and after reversion is not due to the cleaved GFP. (mean ± s.e. of $N = 6$, letters represent significant differences,
ANOVA and Tukey's honest significant difference *post hoc*, $P < 0.05$). Scale bars, 20 μm.

was verified using transient expression, confocal microscopy and
DAPI staining (Fig. 4a–d; Supplementary Fig. 4). Localization of
the GFP control and GFP:NIa-Pro was consistent with previous
experiments with and without aphids (Fig. 4a,b). In contrast,
GFP:NIa-Pro:NLS localized exclusively to the nucleus with and
without aphids (Fig. 4c). For the GFP:NIa-Pro:NES construct,
some GFP:NIa-Pro:NES still localized to the nucleus before aphid
presence, yet in the presence of the aphids there was a greater
number of cells with GFP:NIa-Pro:NES in the vacuoles compared
with plants expressing GFP:NIa-Pro (Fig. 4d).

Next, we allowed aphids to feed and recorded callose depo-
sition on plants transiently expressing the empty vector (free
GFP), GFP:NIa-Pro, GFP:NIa-Pro:NLS and GFP:NIa-Pro:NES.
Consistent with our previous findings, callose induction by aphids
was significantly reduced in the leaves expressing
NIa-Pro compared with free GFP (Fig. 4e)[9]. Interestingly,
aphid-induced callose was not reduced in plants expressing
GFP:NIa-Pro:NLS, whereas in plants expressing GFP:NIa-
Pro:NES, aphid-induced callose deposition was still inhibited
(Fig. 4e). These results suggest that the presence of NIa-Pro
outside of the nucleus is required to decrease plant defenses. We
also recorded aphid fecundity on plants expressing the NIa-Pro
localization mutants and controls. Similar to our previous
findings, aphid fecundity was significantly higher on plants
transiently expressing GFP:NIa-Pro compared with the free GFP
(Fig. 4f; refs 9,22). In contrast, aphids were not more fecund on
plants expressing GFP:NIa-Pro confined to the nucleus as
compared with the control (Fig. 4f; GFP:NIa-Pro:NLS).

**NIa-Pro relocalizes during actual viral infection**. To test whe-
ther NIa-Pro behaves similarly during actual viral infection, we
expressed the GFP:NIa-Pro fusion in the context of the TuMV
genome using an infectious clone. To do this, we used the TuMV
backbone pCambia/TuMV/6k2:GFP[33] and replaced the 6k2:GFP
with GFP:NIa-Pro (Fig. 5a). To verify that the new construct also
increases aphid fecundity similar to the unmodified TuMV/GFP[9],
we performed aphid fecundity experiments with the TuMV/
GFP:NIa-Pro. Consistent with our previous findings, TuMV/
GFP:NIa-Pro also increased aphid fecundity compared with
controls (Fig. 5b; refs 9,22). A western blot analyses was
performed with an antibody specific to the TuMV coat protein
to verify the infectivity of the new mutant, TuMV/GFP:NIa-Pro
(Fig. 5c). We next examined subcellular localization in the viral
context using confocal microscopy. During infection, GFP:NIa-
Pro also localized to the nucleus and cytoplasm without aphids
and relocalized to the vacuole in the presence of the aphid vectors
(Fig. 5d). Quantification of three independent experiments using
TuMV/GFP:NIa-Pro established that aphid presence significantly
induces GFP:NIa-Pro relocalization in the viral context (Fig. 5e).

Next, we wanted to further confirm vacuole localization using
the TuMV/GFP:NIa-Pro infectious clone, vacuole purification
and western blotting. We first purified vacuoles from leaves
expressing the empty vector or TuMV/GFP:NIa-Pro with or
without aphids present. Next, total protein isolated from vacuoles
was analysed by western blot using a GFP antibody. Consistent
with the microscopy work, GFP:NIa-Pro was only found in
vacuoles purified from leaves with aphids present (Fig. 5f). As a

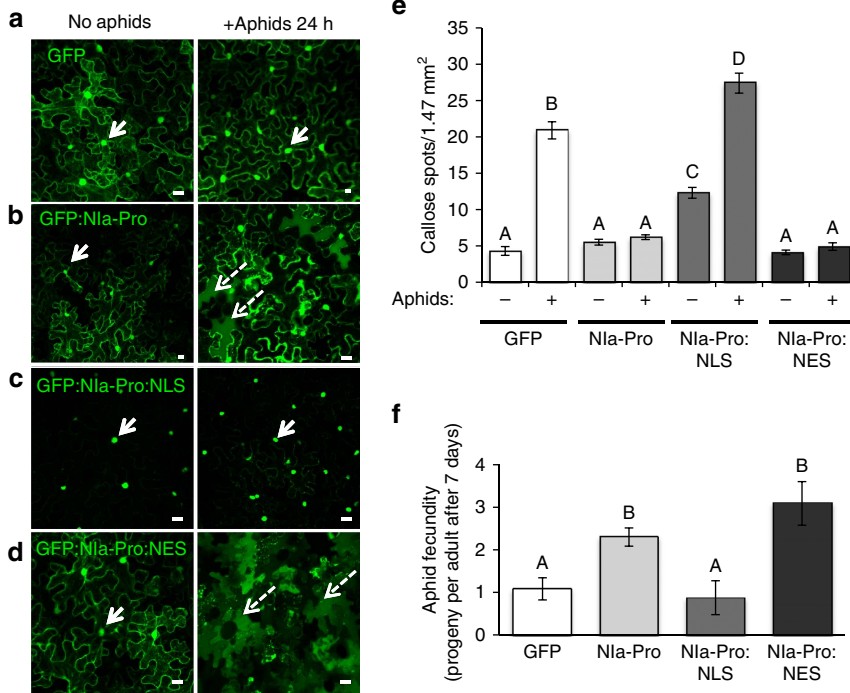

**Figure 4 | Relocalization of NIa-Pro is required to increase aphid fecundity.** Images show confocal projection of *N. benthamiana* leaves agroinfiltrated with (**a**) free GFP, (**b**) GFP:NIa-Pro, (**c**) GFP:NIa-Pro:NLS (NLS, nuclear localization signal) and (**d**) GFP:NIa-Pro:NES (NES, nuclear export signal) with or without aphids. (**e**) Callose deposition in agroinfiltrated *N. benthamiana* leaves with free GFP, GFP:NIa-Pro, GFP:NIa-Pro:NLS and GFP:NIa-Pro:NES with or without aphids infestation. The NLS mutant does not reduce aphid-induced callose accumulation. (**f**) Number of progeny produced by aphids on *N. benthamiana* agroinfiltrated leaves with free GFP, GFP:NIa-Pro, GFP:NIa-Pro:NLS and GFP:NIa-Pro:NES. (mean ± s.e. of $N = 10$ for **e** and $N = 8$–12 for **f**; letters represent significant differences, ANOVA and Tukey's honest significant difference *post hoc*, $P < 0.05$). Results from one of three independent experiments are displayed. Scale bars, 20 μm.

loading control a TuMV-CP antibody was used to confirm equal isolation of vacuoles from different treatments (Fig. 5f). Recently, TuMV particles were found in the vacuoles[34].

**The function of NIa-Pro is conserved.** As all potyvirus genomes encode NIa-Pro, we next wanted to determine whether this is a more generalized phenomenon. To address this, we examined the impact of another potyvirus, *Potato virus Y* (PVY), and expression of its NIa-Pro in host plants on aphid vector performance. Plants infected with PVY increased aphid fecundity compared with controls, similar to TuMV infection (Fig. 6a; refs 9,22). Perhaps, more interesting plants transiently expressing GFP:NIa-Pro[PVY] also increased aphid fecundity compared with aphids feeding on controls (Fig. 6b). Next, we examined NIa-Pro[PVY] localization using GFP and confocal microscopy. Similar to the phenomenon observed with GFP:NIa-Pro[TuMV], GFP:NIa-Pro[PVY] also localized to the nucleus without aphids and to the vacuole with aphids present (Fig. 6c). Quantification of three independent experiments demonstrated that aphid presence significantly modifies GFP:NIa-Pro[PVY] localization (Fig. 6d). This observation suggests NIa-Pro may have a conserved function for the genus *Potyvirus*, representing over 30% of all described plant viruses[19].

**NIa-Pro relocalization is insect dependent.** We next wanted to determine whether GFP:NIa-Pro also relocalizes in the presence of others piercing-sucking insects, or whether it is specific to the aphid vector. To address this, we examined GFP:NIa-Pro relocalization in the presence of two distinct phloem-feeding insects: the beet leafhopper *Circulifer tenellus* and the silverleaf whitefly, *Bemisa tabaci* biotype A. Like aphids,

whiteflies feed by moving their stylets intercellularly on the way to the phloem causing minimal cell damage. However, whiteflies generally make fewer intracellular probes before reaching the phloem and thus feeding is not equivalent to aphids[35]. Leafhoppers, on the other hand, have larger mouthparts compared with aphids and whiteflies. Because of this, leafhoppers generally damage many cells on their intracellular pathway to the phloem[35]. Some relocalization of GFP:NIa-Pro to the vacuole was observed in the presence of leafhoppers and whiteflies (Fig. 6e). However, the number of cells with GFP:NIa-Pro in the vacuole was significantly higher for aphid vectors compared with non-vector insects (aphids: ~60% of cells; leafhoppers and whiteflies: ~30% of cells; Fig. 6e). This suggests relocalization is not correlated with feeding damage, as whiteflies cause less damage and leafhoppers cause more damage during feeding as compared with aphids. Although previous studies have demonstrated that both whiteflies and leafhoppers can transmit viruses to *N. benthamiana* plants[36,37], we next wanted to confirm insects were probing in our experiments. To address this, we collected the leaves that aphids, whiteflies and leafhoppers were confined to and stained them with acid fuchsin. Staining with acid fuchsin allowed us to visualize the saliva sheaths and verify plants were probed[38] (Supplementary Fig. 5).

**NIa-Pro relocalization is host dependent.** Although numerous studies have shown viruses can manipulate plant metabolism and promote vector performance, there is wide variation in response both within and across plant species[39]. Previous studies have shown that NIa-Pro increases aphid performance on *Arabidopsis thaliana* in addition to *N. benthamiana*[9,22], suggesting that this

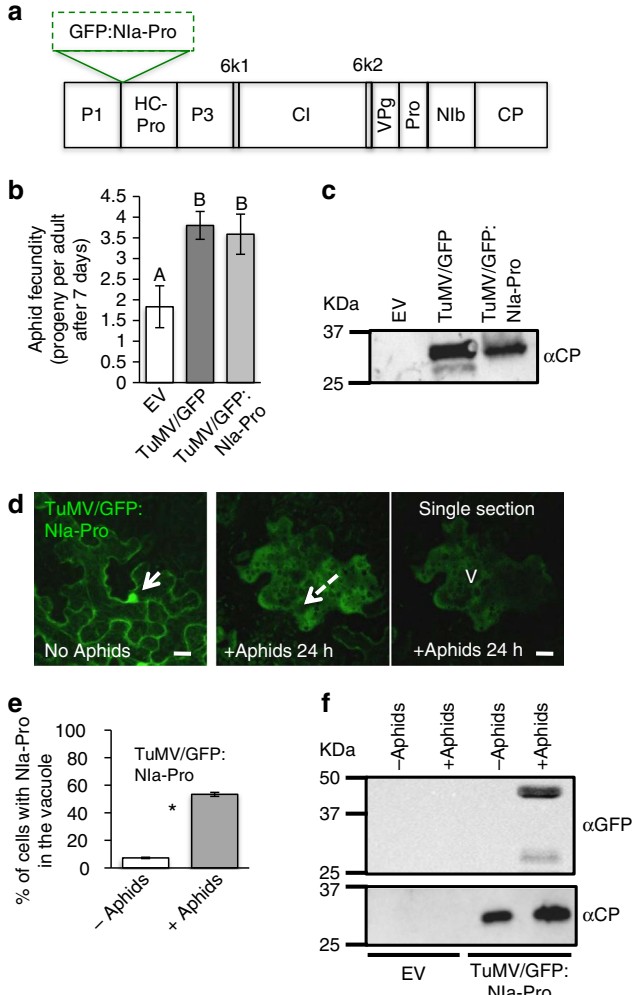

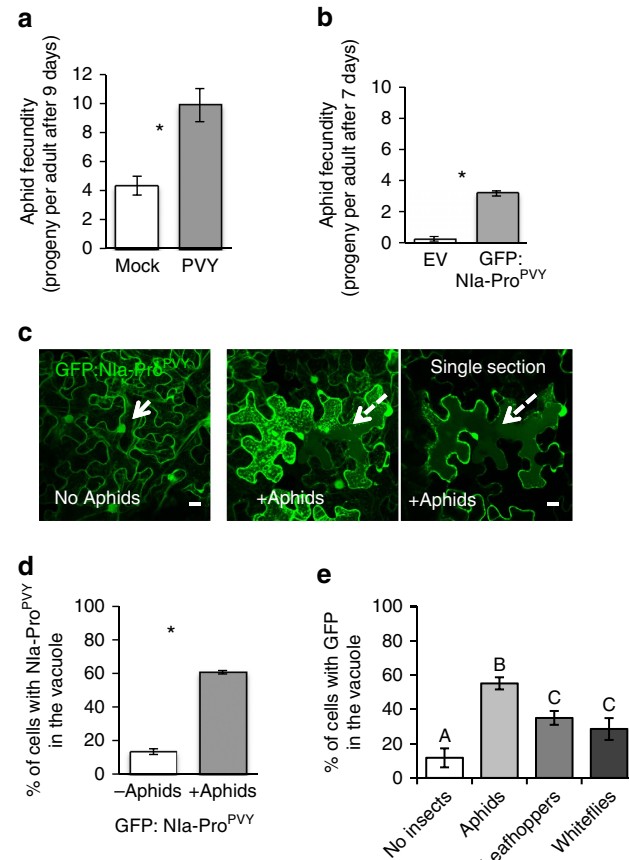

**Figure 5 | NIa-Pro relocalizes to the vacuole in the presence of aphids during virus infection.** (**a**) Schematic representation of the recombinant TuMV-expressing GFP:NIa-Pro. Each rectangle represents individual proteins of the TuMV polyprotein. (**b**) Number of progeny produced by aphids on *N. benthamiana* infiltrated with empty agrobacterium (EV), TuMV/GFP or TuMV/GFP:NIa-Pro. (**c**) The image depicts a western blot from leaf extracts expressing empty vector (EV), TuMV/GFP and TuMV/GFP:NIa-Pro incubated with an antibody that binds to the coat protein (CP) of TuMV. (**d**) Images show confocal projection and a single section (on the right) of *N. benthamiana* leaves agroinfiltrated with TuMV/GFP:NIa-Pro with or without aphids for 24 h. GFP:NIa-Pro relocalizes from the nucleus (white arrow) to the vacuole (V, dashed arrow) in the presence of aphids. (**e**) The percentage of cells with GFP in the vacuole for *N. benthamiana* infected with TuMV/GFP:NIa-Pro with and without aphids. (**f**) The image represents a western blot from vacuole purification derived from leaves expressing EV or TuMV/GFP:NIa-Pro before or after aphid presence and incubated with a GFP antibody and an anti TuMV-CP antibody. The western blot shows that GFP is only visible in the vacuoles from leaves expressing TuMV/GFP:NIa-Pro and after aphid presence. (mean ± s.e. of $N = 12–20$ for **b** and $N = 6$ for **e**; letters and stars represent significant differences, ANOVA and Tukey's honest significant difference *post hoc* for **b** or Student's *t*-test for **e**, $P < 0.05$). Scale bars, 20 µm.

**Figure 6 | The function of NIa-Pro in virus–plant–aphid interactions is conserved.** (**a**) The number of progeny produced by aphids on healthy (mock) or PVY-infected *N. benthamiana*. (**b**) The number of progeny produced by aphids on *N. benthamiana* agroinfiltrated with empty agrobacterium (EV) or expressing GFP:NIa-Pro from PVY. (**c**) Images show confocal projection and single section (on the right) of *N. benthamiana* leaves agroinfiltrated with GFP:NIa-Pro from PVY with or without aphids for 24 h. GFP:NIa-Pro relocalizes from the nucleus (white arrow) to the vacuole (dashed arrow) in the presence of aphids. (**d**) The percentage of cells with GFP in the vacuole for *N. benthamiana* expressing GFP:NIa-Pro from PVY with and without aphids. (**e**) The percentage of cells with GFP:NIa-Pro in the vacuole for each of the treatments: agroinfiltrated *N. benthamiana* with no insects, with aphids, with leafhoppers or with whiteflies for 24 h. (mean ± s.e. of $N = 24–27$ for **a**, $N = 6–9$ for **b**, $N = 6$ for **d** and $N = 3$ for **e**; stars and letters represent significant differences, Student's *t* test, $P < 0.05$ for **a–d**, and ANOVA and Tukey's honest significant difference *post hoc* for **e**, $P < 0.05$ ). Results from one out of two independent experiments are displayed in **a,b**, and from three independent experiments in **d**. Scale bars, 20 µm.

may be a general response across plants. To determine whether NIa-Pro also relocalizes in *Arabidopsis* in the presence of aphids, we generated transgenic plants expressing free GFP or GFP:NIa-Pro as previously described[40]. Consistent with the *N. benthamiana* experiments, after 24 h of aphids infestation

GFP:NIa-Pro was observed in the vacuole, whereas for free GFP there was no relocalization (Fig. 7a).

Next, we examined insect performance on *Nicotiana tabacum* plants infected with TuMV, PVY or expressing the corresponding NIa-Pro. In contrast to *N. benthamiana* and *Arabidopsis* plants[9,22], aphids were not more fecund on infected *N. tabacum* plants compared with controls (Fig. 7b). Furthermore, aphids did not have enhanced fecundity on *N. tabacum* plants expressing GFP:NIa-Pro[TuMV] or GFP:NIa-Pro[PVY] compared with the appropriate controls (Fig. 7b). We next conducted confocal microscopy experiments using transient expression in *N. tabacum* to determine whether GFP:NIa-Pro is still able to relocalize to the vacuole. In contrast to the other host plants, GFP:NIa-Pro[TuMV]

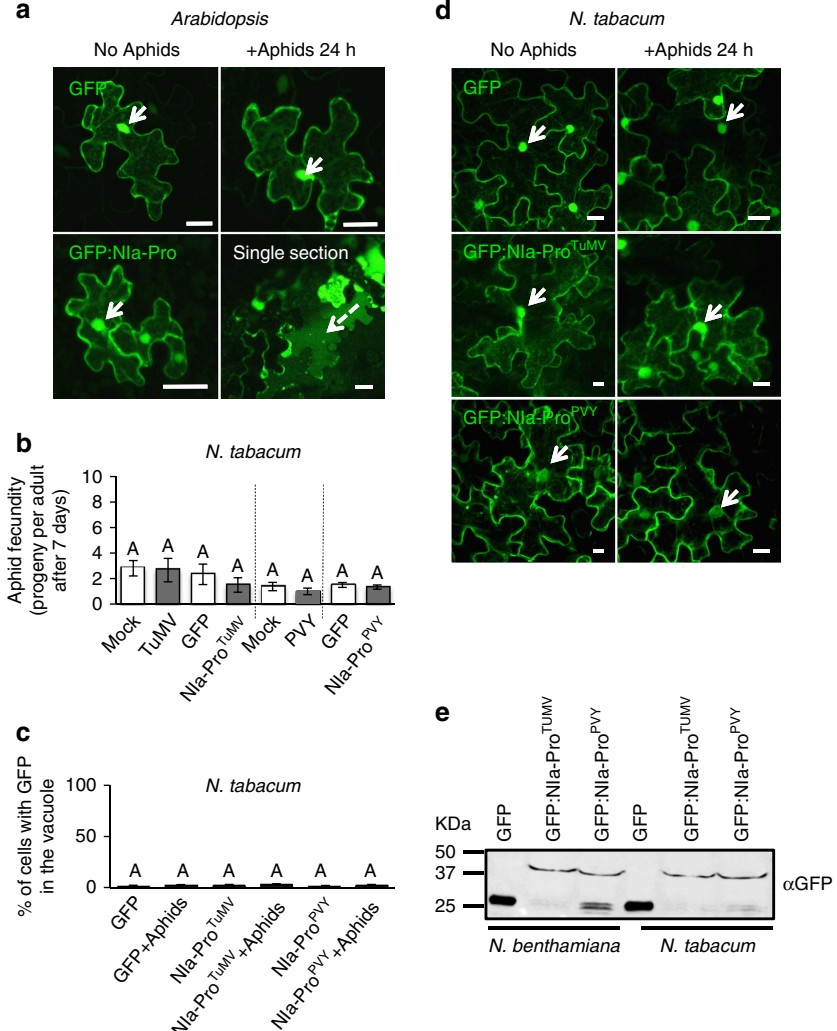

**Figure 7 | NIa-Pro relocalization and impact on aphid fecundity is host plant dependent.** (**a**) Images show confocal projections and a single section of transgenic *A. thaliana* leaves expressing free GFP or GFP:NIa-Pro with or without aphids present. (**b**) Number of progeny produced by aphids on healthy *N. tabacum* or on *N. tabacum* plants inoculated or agroinfiltrated with free GFP, TuMV, PVY, GFP:NIa-Pro$^{TuMV}$ or GFP:NIa-Pro$^{PVY}$. (**c**) The percentage of cells with GFP in the vacuole for *N. tabacum* plants expressing free GFP, GFP:NIa-Pro$^{TuMV}$ and GFP:NIa-Pro$^{PVY}$ with or without aphids. (**d**) Images show confocal projections from **c**. GFP localizes to the nucleus but does not localize to the vacuole during aphid presence in *N. tabacum* plants. (**e**) The image represents a western blot made with leaf extracts from *N. benthamiana* or *N. tabacum* leaves agroinfiltrated with free GFP, GFP:NIa-Pro$^{TuMV}$ and GFP:NIa-Pro$^{PVY}$, and incubated with an GFP antibody. We can see that the expression of free GFP, GFP:NIa-Pro$^{TuMV}$ and GFP:NIa-Pro$^{PVY}$ is similar in the different hosts. Mean ± s.e. of $N = 6$ for **a,c,d** and $N = 8$–12 for **b**; letters represent significant differences, ANOVA and Tukey's honest significant difference *post hoc*, $P < 0.05$. Results from one out of two independent experiments are displayed in **a** and from three independent experiments in **c**. Dashed arrow indicates the vacuole and white arrows indicate the nucleus. Scale bars, 20 µm.

and GFP:NIa-Pro$^{PVY}$ did not relocalize in *N. tabacum* with aphids present (Fig. 7c,d), further supporting the role of relocalization in increased aphid fecundity and changes in plant metabolism. A western blot demonstrated that the expression level was similar for free GFP, GFP:NIa-Pro$^{TuMV}$ or GFP:NIa-Pro$^{PVY}$ in *N. tabacum* and in *N. benthamiana* (Fig. 7e). These observations suggest NIa-Pro must respond to the presence of the vector using a host plant-derived signal.

## Discussion

Our results demonstrate that TuMV initiates host changes in response to vector presence, promoting insect performance and the likelihood of transmission at the most opportune moment. We show that NIa-Pro from TuMV relocalizes to the vacuole in the presence of insect vectors on infected hosts (Figs 1c,2,3a,b

and 5d–f). Further, we demonstrate that relocalization is essential for NIa-Pro's ability to inhibit plant defenses and increase vector fecundity on host plants (Fig. 4e,f). The large number of cells observed with the relocalization phenotype suggests that NIa-Pro responds to a systemic signal that moves throughout the leaf from the cells the aphid come in contact with. Almost twice as much NIa-Pro relocalization was observed in the presence of aphid vectors as compared with non-vector insects (Fig. 6e). Because NIa-Pro relocalization did not correlate with insect damage (that is, whiteflies cause less damage and leafhoppers cause more damage compared with aphids; Fig. 6e), the systemic signal may be only partially due to the physical damage caused during insect probing. This suggests vector-specific factors may also contribute to the systemic signal recognized by NIa-Pro, such as salivary secretions, or other aphid-derived material. Our results also demonstrate that NIa-Pro from PVY, another potyvirus, has the

same properties (Fig. 6), suggesting a conserved role for the protein in vector–host interactions. Finally, we demonstrate that the nature of the host is very important for relocalization of NIa-Pro and the impact of viral infection on host–vector interactions, shedding light on new aspects that may determine viral host range and epidemiology in nature.

Previous studies have reported that animal and plant viruses can alter host physiology and host–vector interactions in ways that are predictive of enhanced transmission[9,22,41–45]. Our results demonstrate viruses can alter host plant physiology in response to vector presence, promoting insect performance and transmission through host plant changes only when needed. The virus must 'recognize' the presence of the vector and respond actively through changes in host physiology, a phenomenon that has not been previously demonstrated for any animal or plant viruses to our knowledge. Recent evidence has demonstrated CaMV responds actively to its aphid vector through structural changes[11,46]. In these studies, aphid-induced changes in CaMV transmission morphs increased the likelihood of transmission by the insect vector[11,46]. Our findings are unique compared with previous studies with CaMV and viral 'recognition' of aphid vectors as CaMV proteins only responded in cells that were directly damaged by the aphids during the probing punctures[11]. Our study may also show a slightly different mechanism, where it is not the virus recognizing the vector and initiating host changes for enhanced transmission but possibly the aphid manipulating the viral protein to suppress aphid defenses. Although the exact mechanisms are still unknown, it may be a 'perceptive insect behaviour' instead of a 'perceptive viral behaviour'.

Aphids move their needle-like mouthpart, the stylet, intercellularly to the phloem and puncture multiple cells before feeding (for review, see ref. 6). Because most plant cells have one large vacuole that occupies up to 90% of the cell volume, aphids may encounter vacuoles often during probing. The vacuole has numerous functions, including storage of metabolic products known to be involved in plant defence against microbial pathogens and herbivores[47–50]. Vacuolar localization of NIa-Pro during insect presence might be important for inhibition of aphid-induced plant signals or defenses known to be stored in the vacuole[50,51]. Previous studies have shown that NIa-Pro has protease activity[52] and DNAse activity[53,54]. As a protease, NIa-Pro could degrade proteins in the vacuole directly involved in plant defence or known to activate other molecule in defence pathways. Recently, a study has shown that TuMV particles and HC-Pro, another TuMV protein, also accumulate in the vacuole[34]. HC-Pro is involved in aphid transmission, although the role of vacuolar localization in transmission remains unknown.

This is also the first evidence that NIa-Pro can localize into the nucleus without VPg. Previous studies have demonstrated that VPg encodes a NLS (Fig. 1a), mediating VPg-NIa-Pro accumulation in the nucleus, although its function in the nucleus remains unclear[20,55]. It was previously suggested that once NIa-Pro is liberated from NIa, it might diffuse into the cytoplasm, as it is only 27 kD (ref. 20). Here we demonstrate that VPg is not essential for nuclear localization. Interestingly, NIa-Pro does not have any known NLSs, suggesting an unknown signal exists or that it uses a different mechanism to move in and out of the nucleus.

Vector-borne pathogens account for >17% of all infectious diseases worldwide, contributing to reductions in agricultural productivity, disrupted ecosystem services and over 1 million deaths each year[56]. Viral pathogens specifically are the aetiologic agents of 5 of the 10 most important vector-borne diseases worldwide, in terms of impact on mortality[56]. Surprisingly, very few studies using animal or plant viruses have examined the impact of viruses in combination with their vectors on hosts. The phenomenon described here not only opens up new areas of research for plant virology but also poses new questions for vector-borne viruses of humans and other animals. For example, mosquito vectors preferentially feed on hens infected with an arbovirus compared with control hens[42] and lambs infected with Rift Valley fever virus are fed on more frequently by the mosquito vectors compared with uninfected animals[43]. These studies suggest host traits that influence transmission dynamics also change in animals systems after infection, but the mechanisms mediating these changes remain unknown. These studies and ours are contributing to a rapidly changing image of virus transmission and in particular the host–virus–vector relationship, which is more complex than expected.

## Methods

**Plants and growth conditions.** *Nicotiana benthamiana* and *Arabidopsis* seeds were originally obtained from Georg Jander (Boyce Thompson Institute, Ithaca, NY). *Nicotiana tabacum* cv. Glurck were obtained from Bryce Falk (University of California, Davis, CA). Plants were grown in growth chambers under controlled conditions (25/20 °C day/night with a photoperiod of 12/12 h day/night).

**Aphids.** Non-viruliferous aphid clones of a tobacco-adapted red strain of *Myzus persicae* were reared under controlled conditions (25/20 °C day/night with a photoperiod of 12/12 h day/night) on *N. tabacum*. Aphids were originally obtained from Georg Jander (Boyce Thompson Institute, Ithaca, NY).

**Virus infection.** TuMV/GFP was propagated from the infectious clone p35:TuMV/GFP[53] by agroinfiltration as described in the 'Protein expression in *Nicotiana* spp' section below. PVY-O was obtained from Stewart Gray at Cornell University, Ithaca, NY[57] and propagated by rub-inoculation. To prepare inoculum, fully infected *N. benthamiana* leaves were collected 3 weeks after inoculation and weighed. Leaves were then ground in two volumes 20 mM phosphate buffer (pH 7.2). For experiments, two leaves per plant were dusted with carborundum (Sigma-Aldrich, St Louis, MO) and rub-inoculated with the virus sap using a cotton-stick applicator. A corresponding set of control plants were dusted with carborundum and mock-inoculated with a cotton-stick applicator that was soaked in uninfected *N. benthamiana* sap in 20 mM phosphate buffer.

**Aphid fecundity.** One apterous adult aphid was placed on the underside of a fully infected or agroinfiltrated *N. benthamiana* leaf and confined in a plastic clip cage (2 cm diameter). After 24 h, all aphids except one nymph were removed. After 7–9 days, depending on the experiment, the progeny of this single nymph were counted to determine fecundity.

**Plasmid constructs.** The constructs were produced using the Gateway cloning kit (Invitrogen, Carlsbad, CA, USA, http://www.lifetechnologies.com) following the manufacturers instructions. In brief, PCR products were amplified using gene-specific primers for each construct flanked by the attB1 and attB2 universal primers (Supplementary Table 1). The PCR products were cloned into the pDONR207 vector using BP clonase and then they were re-cloned into the pSITE:GFP destination vector[58,59] or the pTA7001destination vector[40], using LR clonase. To construct the localization mutants, the NLS from SV40 (PKKKRKV) and the consensus NES (NELALKLAGLDINK) sequence were added in the primers[31,32].

The TuMV/GFP:NIa-Pro construct was created by amplifying a PCR product with pSITE:GFP:NIa-Pro as the template using oligonucleotides containing a SacII restriction site (SacII-eGFP-F and NIa-Pro-SacII-R; Supplementary Table 1). The PCR product was digested with the SacII restriction endonuclease and inserted into the corresponding restriction endonuclease site in TuMV/6K2:GFP after removal of 6K2:GFP and dephosphorylation[33,60].

**Protein expression in *Nicotiana* spp.** Constructs containing the coding sequence for the proteins fused to *gfp* were introduced into *Agrobacterium tumefaciens* GV3101 by heat shock and selected on LB plus 10 μg ml$^{-1}$ of rifampicin and 50 μg ml$^{-1}$ of spectinomycin for the pSITE plasmid or plus 10 μg ml$^{-1}$ of rifampicin and 50 μg ml$^{-1}$ of kanamycin for pCAMBIA and spRFP:AFVY. One fresh colony was selected and grown overnight in liquid culture. The pellet of the culture was resuspended in 10 mM MES pH 5.85; 10 mM MgCl$_2$ and 150 μM acetosyringone and left at room temperature for 2 h. The solution was then diluted to an optical density of 0.1 at 600 nm for transient expression experiments and at 0.3 for the infectious clones. Single leaves from 3-week-old *N. benthamiana* or *N. tabacum* plants were then agroinfiltrated with the solution. For *N. benthamiana*, entire leaves were infiltrated. For *N. tabacum*, a circle of ∼5 cm diameter was infiltrated for each leaf. Plants were used in experiments 2 days after infiltration.

Expression was verified by microscopy, PCR and/or western blot as previously described[9].

**Transgenic *Arabidopsis*.** The coding sequence from GFP or GFP:NIa-Pro was inserted into pTA7001 plasmid, containing the glucocorticoid-inducible system[40] as described above. Wild-type Columbia (Col-0) *Arabidopsis* plants were transformed using the floral dip method as previously described[61]. The seeds were then plated on MS plate plus 20 µg ml$^{-1}$ of hygromycin. Resistant seedlings were selected after 2 weeks and moved to pots with soil. One month later, the plants were sprayed with 3 µM of dexamethasone and 24 h after plants were used in microscopy experiments with and without aphids.

**Insect presence experiments.** Approximately 50 aphids, 100 whiteflies or 50 leafhoppers were confined in a plastic clip cage with a 2 cm diameter and allowed to feed for 2, 6 or 24 h on the underside of a *N. benthamiana* or *N. tabacum* leaf, 2 days post agroinfiltration. The number of insects used for each species was selected to completely cover the surface of the leaf exposed in each cage. A plastic clip cage without insects was used as a control in all experiments. At least one leaf from two different plants was used for each treatment in each experiment and each experiment was repeated three times.

**Stains.** Two days after infiltration or immediately after aphid treatment, leaves were cut into small pieces and emerged in 10 µg ml$^{-1}$ of DAPI to stain the nucleus (Sigma-Aldrich, St. Louis, MO) or with 10 µM of SNARF-1 (Invitrogen, Carlsbad, CA, USA, http://www.lifetechnologies.com) to stain the vacuole for 2 h at room temperature.

**Salivary sheath staining.** Insects were caged to *N. benthamiana* leaves as described above. After 24 h, insects were removed and leaves were cleared in 70% ethanol for at least 48 h. Following clearing, leaves were soaked in an acid fuchsin solution (0.035% in acetic acid:water, 1:3 V) for 2 min at room temperature[38] and then immediately rinsed with water. Next leaves were mounted on a glass slide in 50% glycerol and analysed directly using a light microscope.

**Microscopy.** Leaves were cut into small pieces and placed on glass slides with a drop of perfluorodecalin (Sigma-Aldrich, St Louis, MO) and observed with a Leica TCS SP5 confocal laser scanning microscope system (Leica Micro- systems, Bannockburn, IL, USA). GFP fluorescence was detected with excitation at 488 nm and emission capture at 500–530 nm. The DAPI was excited with a 405 nm laser, and emission was collected from 405–500 nm. SNARF-1 was excited at 514 nm and emission was collected from 600–700 nm. spRFP:AFVY was excited at 532 nm and emission was collected from 550–700 nm. Images were captured at 2,361 µm intervals using a × 20 objective Final figures were prepared using ImageJ software (imagej.nih.gov/ij/).

**NIa-Pro localization quantification.** Five microscopy fields were randomly selected for each leaf and visualized. For each field, the number of cells showing GFP inside the nucleus or inside the vacuole was quantified. Two leaves from separate plants were examined for each treatment and three independent experiments were conducted.

**Vacuole purification.** The vacuole isolation was performed as previously described[32]. Briefly, *N. benthamiana* leaves agroinfiltrated with TuMV/GFP:NIa-Pro or the empty vector were sliced into 1 mm stripes with a razor blade. The cut leaves were then placed in protoplastation solution (0.4 M mannitol, 20 mM Mes, pH 5.7. 20 mM KCl, 1.5% (w/v) Cellulase R-10, 0.2% (w/v) Macerozyme R-10, 0.1% (w/v) BSA, 10 mM CaCl$_2$). Vacuum was applied for 20 min followed by dark incubation for 3.5 h at room temperature. The protoplasts were then filtered with a miracloth filter, and centrifuged at 80g for 5 min. Protoplasts were washed two times in washing buffer (0.4 M mannitol, 20 mM Mes, pH 5.7), and then resuspended in 10 ml of prewarmed (37 °C) lysis buffer (0.2 M mannitol, 10% (w/v) Ficoll, 10 mM EDTA, pH 8.0, 5 mM sodium phosphate, pH 8.0). After 5 min of incubation, 5 ml of the solution was overlayed with 3 ml 4% (w/v) Ficoll solution and 1 ml ice-cold vacuole buffer (0.2 M mannitol, 2 mM EDTA, pH 8.0, 5 mM sodium phosphate, pH 7.5). The gradient was centrifuged at 1,500g for 20 min at 10 °C, and the vacuoles were collected from the interface between 0 and 4% Ficoll. The purity and quality of the protoplast isolations and of vacuole isolations were accessed using light microscopy (Supplementary Fig. 6).

**Western blotting.** Two days post agroinfiltration, the leaves were crushed in a lysis buffer (50 mM Tris pH 7.5, 300 mM NaCl, 1 mM PMSF, 1% Cocktail antiprotease) and then boiled in Laemmeli buffer[62]. The plant extract was then centrifuged for 5 min at 8,000g. The supernatant was fractionated by a 12% SDS–polyacrylamide gel electrophoresis gel under reducing conditions and transferred to a nitrocellulose membrane using a transfer apparatus according to the manufacturer's protocols (Bio-Rad, Hercules, CA). After incubation with 5% nonfat milk in PBST (137 mM NaCl, 2.7 mM KCl, 8 mM Na$_2$HPO$_4$, 2 mM KH$_2$PO$_4$, 0.3% Tween 20) for 30 min, the membrane was incubated with an antibody against GFP (1:2,000 dilution) for 2 h at room temperature. The GFP antibody was already conjugated to the horseradish peroxidase (Anti-GFP-HRP, http://www.miltenyibiotec.com, #130-091-833). For a loading control, the membrane was also incubated with an antibody specific to the coat protein of TuMV (provided by J-F Laliberté) (1:1,000 dilution) for 2 h at room temperature and then with a horseradish peroxidase-conjugated goat anti-rabbit antibody (Bio-Rad, #1706515, 1:10,000). Blots were washed with PBST three times for 10 min each and developed with an enhanced chemiluminescence system according to the manufacturer (Bio-Rad, Hercules, CA). All uncropped gel/blot images are available in Supplementary Figs 7 and 8.

**Statistical analysis.** The statistical analyses were performed using JMP 8 software (SAS Institute, Cary, NC, USA) and data were analysed by Student's *t*-tests or analysis of variance followed by a Tukey's honest significant difference *post hoc* test. All experiments were repeated at least three times. For fecundity, each experiment had at least 12 experimental units per repetition depending of the experiment. For localization experiments, at least two plants were used for each treatment and three repetitions were performed.

**Data availability.** The authors declare that all data supporting the findings of this study are available in the manuscript and its Supplementary Information files or are available from the corresponding author upon request.

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

## Acknowledgements

We thank Georgia Drakakaki for advice on vacuole localization experiments and use of the SNARF-1 dye from her lab. We thank Jean-François Laliberté for providing the original pCambia/TuMV/6k2:GFP construct from his publications and for the TuMV-CP antibody. We thank Lorenzo Frigerio for the spRFP:AFVY construct. We thank Bob Gilbertson for providing beet leafhoppers and Bryce Falk for providing whiteflies. This publication was supported by the United States Department of Agriculture award 2015-04607 and University of California start-up funds to C.L.C., and by a grant from the US National Science Foundation, IOS-1121788 to S.A.W.

## Author contributions

C.L.C. conceived the project. A.B and C.L.C. designed the research. A.B., A.C., C.Y. and C.L.C. performed research. A.B., A.C. and C.L.C. analysed the data. C.Y. and S.A.W. provided input on experimental design. A.B. and C.L.C. wrote the article with contributions of all the authors.

## Additional information

**Competing financial interests:** The authors declare no competing financial interests.

