## [Peer Review File · Nature Communications]

Reviewers' comments:

Please note that reviewer #2 has organized their report according to our template which asks reviewers to comment as follows:

A Summary of the key results

B Originality and interest: if not novel, please give references

C Data & methodology: validity of approach, quality of data, quality of presentation

D Appropriate use of statistics and treatment of uncertainties

E Conclusions: robustness, validity, reliability

F Suggested improvements: experiments, data for possible revision

G References: appropriate credit to previous work?

H Clarity and context: lucidity of abstract/summary, appropriateness of abstract, introduction and conclusions

Reviewer #1 (Remarks to the Author):

The laboratory of Clare Casteel has previously shown that aphids feeding on plants infected by turnip mosaic virus (TuMV) have an increased fitness. The group has demonstrated that this increased fitness resulted from the sole production of the viral protein NIa-Pro. This is a very interesting but peculiar observation as it is difficult to conceive how a single protein could induce such a complex behavior in insects, even more so when there is no viral infection.

In this submitted manuscript, the authors have tried to go a step further in trying to find an explanation for the role of NIa-Pro. They suggested that the viral protein is relocated from the nucleus to the central vacuole upon insect feeding. I was not, however, convinced of their conclusion, quite to the contrary. Furthermore, I found that their data provided only incremental information on how NIa-Pro production increases aphid fecundity.

Here is a list of issues that substantiate my evaluation:

P. 5: "GFP:NIa-Pro localizes to the nucleus in *N. benthamiana*" and P. 6 : " ... GFP:NIa-Pro is ... too large to diffuse into the nucleus ..." The title of this section is not appropriate. It gives the impression that the viral protein is found only in the nucleus, which is not the case as it is also found in the cytoplasm. The assertion that NIa-Pro is a nuclear protein is central to the authors' claim that it relocates from the nucleus to the vacuole during insect feeding. First, it would be important to show that NIa-Pro (not the precursor VPg-NIa-Pro) is found in the nucleus during infection, which to my knowledge has not been done. To state that GFP:NIa-Pro is too large to simply diffuse in the nucleus is not enough. It is important to show that the nuclear localization of the viral protein is an active process.

P. 6: "Around 60% of the cells ... exhibit a relocation of NIa-Pro ..." Nuclei are often found deep in epidermal cells, and are not necessarily seen during confocal microscopy. For example, looking at Fig. 2, several epidermal cells are imaged but only one nucleus is seen for each panel. Consequently, DAPI staining must be shown to prove that no NIa-Pro is

found in nuclei.

P. 6: "... GFP:NIa-Pro colocalizes perfectly with the vacuole ..." Since the vacuole takes much of the volume of the cell, the cytoplasm is pushed to a very thin space between the tonoplast and plasma membrane. Since the thickness of the optical slice is not specified in the Methods section (often 1 μm), what one sees may represent soluble GFP:NIa-Pro located above the SNARF-1 signal, giving the illusion of colocalization. Moreover, what is the meaning of the punctates seen with the vacuole marker? Since NIa-Pro localization in this organelle is central to the claims of the authors, the confocal data needs to be corroborated by other means, for example vacuole purification and western blotting for the viral protein.

P. 7: "Relocalization of GFP:NIa-Pro is required to increase the aphid fecundity." This conclusion is false, since GFP:NIa-Pro:NES, which does not "relocalize from the nucleus, is effective for aphid fecundity. What it appears to me is that NIa-Pro has to be in the cytoplasm to be effective, and that the claim that it has to "relocalize from the nucleus" is irrelevant.

minor points:

P. 5 : I did not find the data indicating that "host plant changes mediating increased aphid fecundity are conferred by the NIa-Pro protein and not by its RNA sequence." worth showing.

Fig. 3: The authors need to comment on the existence the "shadow" band in the NIa-Pro lane.

P. 6: "... aphid presence induced a relocalization of GFP:NIa-Pro in the cell ..." Since GFP:NIa-Pro is already present in the cytoplasm, perhaps it is better to write "aphid presence precluded GFP:NIa-Pro to enter the nucleus".

P. 8: "... GFP:NIa-Pro relocalization (to the vacuole) in the viral context ..." Vacuoles need to be purified to make this statement.

Reviewer #2 (Remarks to the Author):

A This manuscript describes the relocation of a virus protein upon interaction of the host plant with the virus vector. It also showed that this relocalization improves fertility of the insect.

B The results are novel, i.e. they not only show that a virus protein attracts insects and improves transmission, but also that the insect is rewarded.

C The approach is valid, the data are clear, necessary controls have been made.

D not relevant here

E see C

F maybe proof-read the MS more carefully, for instance change

"Due to their immobility, many viruses rely on other organisms for transmission, including ~80% of all plant infecting viruses" to

"Due to their immobility, many viruses, including ~80% of all plant infecting viruses, rely on other organisms for transmission."

change "viruses for vegetables" to "viruses of vegetables"

change "moving intercellularly the phloem" to "moving intercellularly into the phloem"

change "infectious diseases worldwide" to "infectious diseases of animals worldwide"

etc, etc.

G references are appropriate

H The text is easy to read and the parts are appropriate.

Reviewer #3 (Remarks to the Author):

The manuscript describes experiments where a GFP-tagged Viral Nia-Pro expressed in epidermal cells is observed to change localisation and move into the vacuole on aphid colonisation of the leaf and then return (relocalise) to the original cellular localisations after aphids are removed.

This is a new observation and is of broad interest but the authors should address the questions below to strengthen the claims and the experimental evidence.

Another point to consider is that Potyviruses are transmitted in a non persistent manner after brief probes into the cells and probing is more important than feeding in virus transmission. There is some evidence that short probes (1-2 min) result in more efficient transmission.

Therefore, the results could be considered differently in that it's not virus recognising the vector for virus benefit and initiating host changes but it's the aphid manipulating virus protein to help suppress aphid defence response. Thus not perceptive viral behaviour but perceptive insect behaviour as authors suggest page 11 para 2 lines 6-10

Specific comments

Final sentence in introduction seems extraneous - it does not follow the rest of the paragraph

GFP- Nia-Pro localisation not only in nucleus can be seen in cytoplasm and small round bodies; this need fuller description e.g. Fig 2.. Images seem to show cytoplasm, spheres, spots at periphery. Fig 2h reversion, can you be sure this is reversion? By marking the cell and returning to image the same cell 2 h after removal of aphids it would be possible to see if it was truly reversion or whether cell death occurred.

Is the relocalisation due to feeding or just probing (presumably aphids must feed as they reproduce). Earlier time points might allow to make a decision eg at 6 h not fully relocalised (fig 3)

Fig 3b. +aphids 6 h the cytoplasm seems to be rounding off and disrupted in these cells, GFP-Nia-Pro seems to be still localised to nucleus. Are these cells still viable?

Fig 3c. +aphids24h to counter the suggestion that these cells could be dead and auto fluorescing; images should be presented by focus on the top of the cell and take a time series of a single Z plane to show cytoplasmic streaming, also record. Spectrum across the vacuole and confirm spectrum is GFP (not auto fluorescence)

Fig 4b see fig 3c comments

Fig5 DAPI stain of control -aphid treatments should be included to verify nuclear localisation

Fig 6c This image would be more convincing if include DAPI and vacuole stains to confirm localications

Page 7. Fig 5e Amend last sentence since aphid fecundity similar statistical significance on NIa-Pro and NIa-Pro-NES

The last para of discussion about animal-virus systems is speculative.

Reviewer #4 (Remarks to the Author):

Due to immobility of plant, around 75% of 1100 plant virus species are piercing-sucking insect transmitted. In this study, Casteel et al. investigated the interaction between a viral protease and insect vector. They provided new evidences to show that the expression and subcellular re-localization of a Potyviral protein NIa-Pro benefits aphid. I like reading this paper, most of it is very well written and a good work was done. This study further reinforced the concept of "perceptive behaviours" proposed by Martiniere et al. 2013 and presents a novel angle to understand complex virus-vector-plant tripartite interactions.

However my biggest concern is the methodology they used. quality of data Most of data are based on agroinfiltration-a transient expression system in plant. It lacks validity of approach. Agroinfiltration can be a good fast test method to provide preliminary results which have to be further proven in stable transgenic lines. Data quality is not good. Results based on stable lines can be much easier to be repeatable among researchers or labs when compared with agroinfiltration method. Without supports from stable transgenic lines all statements are not solid enough and therefore additional vital experiments are essential to make conclusive claims. The authors have generated and used NIa-Pro expression Arabidopsis stable lines in recent publications, for example Casteel et al. Plant Physiology 2015.

Second concern is the method to determine the subcellular localization of vacuole., which I will explain in detail below.

Without these experiments the importance is diminished and the appropriateness for Nature

Communications will be weakened. In addition, essential controls and information are missing in some experiments (see specifics below).

Specifics

Major:

The authors claimed that NIa-Pro can shuttle from the nucleus to the vacuole when the presence of the aphid vectors during infection. The method they only used is the SNARF-1, Dextran-tagged seminaphthorhodafluor-1, a chemical used to stain vacuole of animal cell. The only supporting data shown in Fig. 4 is of low quality. I am far from being convinced at least. I would strongly suggest the authors reading more papers in area of plant cell biology. Actually, plant researchers would like to use fluorescent protein fused with vacuole protein (FP-probes) to mark organelles. You may use TIP1;1-YFP fusion to mark the tonoplast, spRFP-AFVY fusion to mark the lumen.

Second, "vacuoles" shown in Fig. 4 are something strange. Normally for a mature tobacco leaf mesophyl cell the central vacuole is occupying up to 90% of a cell's volume. Meristematic cells contain numerous small provacuoles (resembling animal lysosomes) that arise from fusion of trans-Golgi-derived vesicles (Marty, F. 1999. Plant vacuoles. Plant Cell). The authors need to be very caution to claim NIa-Pro relocalization in vacuoles.

Third, they claimed "The virus must somehow "recognize" the presence of the vector and respond actively". Since they used "recognize", they would like to test with various aphid types effect on this recognize, for example using nonvector aphid or even whitefly as a control.

Minor:

Materials and methods part

Page 13 add *Nicotiana tabacum* cultivar name or specific accession number.

Page 14 line 4 The detailed amino acids for NLS and NES used in the research.

line 9-10. Restrict enzyme sites should be correctly written, SacII

Line 14 GFP should in italic.

Line 16 pCAMBIA not pCambia

Line 19-20. Why not a specific OD at 600 nm, The agrobacterium density from 0.1-0.3 can result in big difference in protein expression when do agroinfiltration.

Nature Communications Reviewer #1 (Remarks to the Author):

The laboratory of Clare Casteel has previously shown that aphids feeding on plants infected by turnip mosaic virus (TuMV) have an increased fitness. The group has demonstrated that this increased fitness resulted from the sole production of the viral protein NIa-Pro. This is a very interesting but peculiar observation as it is difficult to conceive how a single protein could induce such a complex behavior in insects, even more so when there is no viral infection.

In this submitted manuscript, the authors have tried to go a step further in trying to find an explanation for the role of NIa-Pro. They suggested that the viral protein is relocated from the nucleus to the central vacuole upon insect feeding. I was not, however, convinced of their conclusion, quite to the contrary. Furthermore, I found that their data provided only incremental information on how NIa-Pro production increases aphid fecundity.

Here is a list of issues that substantiate my evaluation:

P. 5: "GFP:NIa-Pro localizes to the nucleus in *N. benthamiana*" and P. 6 : " ... GFP:NIa-Pro is ... too large to diffuse into the nucleus ..." The title of this section is not appropriate. It gives the impression that the viral protein is found only in the nucleus, which is not the case as it is also found in the cytoplasm. The assertion that NIa-Pro is a nuclear protein is central to the authors' claim that it relocates from the nucleus to the vacuole during insect feeding. First, it would be important to show that NIa-Pro (not the precursor VPg-NIa-Pro) is found in the nucleus during infection, which to my knowledge has not been done. To state that GFP:NIa-Pro is too large to simply diffuse in the nucleus is not enough. It is important to show that the nuclear localization of the viral protein is an active process.

The reviewer is correct NIa-Pro is found in the nucleus and the cytoplasm. We changed the title and rewrote this section to make it clearer. We agree that it is important to show NIa-pro is found in the nucleus during actual infection and we did this in the original manuscript (See Fig. 7c). We also did this for NIa-Pro in isolation of viral infection and in isolation of VPg (See Fig. 1 and 3). This was done using similar method as Restrepo et al., 1990 with GFP instead of GUS as the reporter.

P. 6: "Around 60% of the cells ... exhibit a relocalization of NIa-Pro ..." Nuclei are often found deep in epidermal cells, and are not necessarily seen during confocal microscopy. For example, looking at Fig. 2, several epidermal cells are imaged but only one nucleus is seen for each panel. Consequently, DAPI staining must be shown to prove that no NIa-Pro is found in nuclei.

We changed the text to clarify that we are referring to 60% of the *visible* cells. We do not state that there is no NIa-Pro in the nucleus after relocalization. Our data actually demonstrates NIa-pro is in the nucleus *and* in the vacuole during the relocalization phenotype (Fig. 3b). However we agree, additional images of the DAPI staining would

be beneficial. We added the requested images with Nla-Pro plus aphids with DAPI staining (Fig. 1c).

P. 6: "... GFP:Nla-Pro colocalizes perfectly with the vacuole ..." Since the vacuole takes much of the volume of the cell, the cytoplasm is pushed to a very thin space between the tonoplast and plasma membrane. Since the thickness of the optical slice is not specified in the Methods section (often 1 μm), what one sees may represent soluble GFP:Nla-Pro located above the SNARF-1 signal, giving the illusion of colocalization. Moreover, what is the meaning of the punctates seen with the vacuole marker? Since Nla-Pro localization in this organelle is central to the claims of the authors, the confocal data needs to be corroborated by other means, for example vacuole purification and western blotting for the viral protein.

We agree that the original images of SNARF-1 staining could be better. We repeated the experiments and included better images demonstrating colocalization. We also included additional information on the microscopy methods as suggested by the reviewer (ie. optical slice). To further demonstrate vacuole localization we performed a second set of microscopy experiments with spRFP-AFVY another vacuole lumen marker. Images from this experiment were also included in the manuscript (Fig. 2). Finally as suggested by the reviewer, we also purified vacuoles from control and TuMV/GFP:Nla-Pro infected plants with and without aphid feeding. Proteins were extracted from the vacuoles and incubated with anti-GFP with western blotting (Fig.5f). Nla-Pro:GFP was only detected in the vacuoles with Nla-pro and in the presence of aphids. Thank you for the suggesting this experiment! It was a good idea!

P. 7: "Relocalization of GFP:Nla-Pro is required to increase the aphid fecundity." This conclusion is false, since GFP:Nla-Pro:NES, which does not "relocalize from the nucleus, is effective for aphid fecundity. What it appears to me is that Nla-Pro has to be in the cytoplasm to be effective, and that the claim that it has to "relocalize from the nucleus" is irrelevant.

GFP:Nla-Pro:NES does relocalizes from the nucleus and aphid fecundity is still increased (Fig.4). The mutant has a phenotype similar to the wt but with a greater amount of localization in the vacuole in the presence of aphids (Fig.4d). We clarified the wording of this in the text.

minor points:

P. 5 : I did not find the data indicating that "host plant changes mediating increased aphid fecundity are conferred by the Nla-Pro protein and not by its RNA sequence." worth showing.

We thought this was important to demonstrate. We have moved this figure to the supplementary data as suggested.

Fig. 3: The authors need to comment on the existence the "shadow" band in the Nla-Pro lane.

The shadow band likely represents a small amount of GFP that has been cleaved from Nia-pro. This is a very common observation when working with GFP labeled proteins over time. However most of the GFP is not cleaved and in other experiments you can see no cleavage GFP (Fig 3). We added a comment in the text to explain this.

P. 6: "... aphid presence induced a relocalization of GFP:Nia-Pro in the cell ..." Since GFP:Nia-Pro is already present in the cytoplasm, perhaps it is better to write "aphid presence precluded GFP:Nia-Pro to enter the nucleus".

GFP:Nia-Pro can still be observed in the nucleus when it colocalizes with the vacuole in certain cells. We cannot speculate if the relocalization of GFP:Nia-Pro comes from the cytoplasm or the nucleus or both during aphid presence, but we can say that there is a relocalization of GFP:Nia-Pro in the cell when aphids are present.

P. 8: "... GFP:Nia-Pro relocalization (to the vacuole) in the viral context ..." Vacuoles need to be purified to make this statement.

We performed vacuole purifications and western blotting as suggested (Fig.5f). Thank you for this recommendation!

Reviewer #2 (Remarks to the Author):

A This manuscript describes the relocation of a virus protein upon interaction of the host plant with the virus vector. It also showed that this relocalizations improves fertility of the insect.

B The results are novel, i.e. they not only show that a virus protein attracts insects and improves transmission, but also that the insect is rewarded.

C The approach is valid, the data are clear, necessary controls have been made.

D not relevant here

E see C

F maybe proof-read the MS more carefully, for instance change

"Due to their immobility, many viruses rely on other organisms for transmission, including ~80% of all plant infecting viruses" to

"Due to their immobility, many viruses, including ~80% of all plant infecting viruses, rely on other organisms for transmission."

The text was changed as suggested.

change "viruses for vegetables" to "viruses of vegetables"

The text was changed as suggested.

change "moving intercellularly the phloem" to "moving intercellularly into the phloem"

The text was changed as suggested.

change "infectious diseases worldwide" to "infectious diseases of animals worldwide"

The text was changed as suggested.

etc, etc.

The manuscript was carefully proof-read and edited again. We also had a second native English speaker read the manuscript to catch any final mistakes. Thank you for taking time to make these editing suggestions.

G references are appropriate

H The text is easy to read and the parts are appropriate.

We proof-read the manuscript and made the modifications that you suggested.

Reviewer #3 (Remarks to the Author):

The manuscript describes experiments where a GFP-tagged Viral Nia-Pro expressed in epidermal cells is observed to change localisation and move into the vacuole on aphid colonisation of the leaf and then return (relocalise) to the original cellular localisations after aphids are removed.

This is a new observation and is of broad interest but the authors should address the questions below to strengthen the claims and the experimental evidence.

Another point to consider is that Potyviruses are transmitted in a non persistent manner after brief probes into the cells and probing is more important than feeding in virus transmission. There is some evidence that short probes (1-2 min) result in more efficient transmission.

Therefore, the results could be considered differently in that it's not virus recognising the vector for virus benefit and initiating host changes but it's the aphid manipulating virus protein to help suppress aphid defence response. Thus not perceptive viral behaviour but perceptive insect behaviour as authors suggest page 11 para 2 lines 6-10

Good point! We rewrote the discussion section to consider the suggested point.

Specific comments

Final sentence in introduction seems extraneous - it does not follow the rest of the paragraph

We removed the final sentence.

GFP- Nia-Pro localization not only in nucleus can be seen in cytoplasm and small round bodies; this need fuller description e.g. Fig 2. Images seem to show cytoplasm, spheres, spots at periphery. Fig 2h reversion, can you be sure this is reversion? By marking the cell and returning to image the same cell 2 h after removal of aphids it would be possible to see if it was truly reversion or whether cell death occurred.

For all the microscopy we have to destructively sample for each treatment and cannot go back to the original cell. We do not know if Nia-Pro is moving between locations or being synthesized at different locations. However, our data does support a reversion to the original phenotype after aphids are removed (Original phenotype = nuclear localization). We changed the title and the description of the Fig. 3 to make this clearer. The western blot in Fig 3c shows that the GFP:NiaPro is still intact during the reversion and cells don't show plasmolysis, a phenotype of cell death/the hypersensitive response (Fig 1 ; Supplemental Fig. 2).

Is the relocalisation due to feeding or just probing (presumably aphids must feed as they reproduce). Earlier time points might allow to make a decision eg at 6 h not fully relocalised (fig 3)

This is a good question but we don't know the answer yet. We added new images at 2h after aphids we placed as suggested. We didn't see relocalization 2 hr after aphids were placed (Fig.3b), suggesting probing is not enough to induce relocalization. However we cannot discount the possibility that probing may still induce a delayed signal. We can not separate probing from feeding at this point and choose not to speculate.

Fig 3b. +aphids 6 h the cytoplasm seems to be rounding off and disrupted in these cells, GFP-Nia-Pro seems to be still localised to nucleus. Are these cells still viable?

This phenotype observed is appears to be prevacuolar compartments. We added information to the text and citations to clarify the observations to the reader.

Fig 3c. +aphids 24h to counter the suggestion that these cells could be dead and auto fluorescing; images should be presented by focus on the top of the cell and take a time series of a single Z plane to show cytoplasmic streaming, also record. Spectrum across the vacuole and confirm spectrum is GFP (not auto fluorescence)

Fig 4b see fig 3c comments

We added a time series to show that the cells with the phenotype are still alive

(Supplemental Figure 2). Thanks for this suggestion - the vacuole localization phenotype does not appear to be autofluorescence. We also contacted the microscope core and the microscope company and the microscope software company to determine how to add the requested graph of excitation/emission spectrum across the vacuole. The lab core and both companies did not know to create the requested graph with our current microscope and software.

Fig5 DAPI stain of control (No aphid) treatments should be included to verify nuclear localisation

The requested images were added (Supplemental Figure 3).

Fig 6c This image would be more convincing if include DAPI and vacuole stains to confirm localisations

To further confirm vacuole localizations we conducted additional microscopy work with additional stains as suggested (Fig. 2) and we purified the vacuoles and conducted western blot analysis to further verify vacuole localization of Nia-Pro during viral infection (Fig.5f).

Page 7. Fig 5e Amend last sentence since aphid fecundity similar statistical significance on NIa-Pro and NIa-Pro-NES

We modified the last sentence as suggested.

The last para of discussion about animal-virus systems is speculative.

We modified the last part of the discussion to soften our speculation.

Reviewer #4 (Remarks to the Author):

Due to immobility of plant, around 75% of 1100 plant virus species are piercing-sucking insect transmitted. In this study, Casteel et al. investigated the interaction between a viral protease and insect vector. They provided new evidences to show that the expression and subcellular re-localization of a Potyviral protein NIa-Pro benefits aphid. I like reading this paper, most of it is very well written and a good work was done. This study further reinforced the concept of "perceptive behaviours" proposed by Martiniere et al. 2013 and presents a novel angle to understand complex virus-vector-plant tripartite interactions.

However my biggest concern is the methodology they used. quality of data Most of data are based on agroinfiltration-a transient expression system in plant. It lacks validity of approach. Agroinfiltration can be a good fast test method to provide preliminary results which have to be further proven in stable transgenic lines. Data quality is not good. Results based on stable lines can be much easier to be repeatable among researchers

or labs when compared with agroinfiltration method. Without supports from stable transgenic lines all statements are not solid enough and therefore additional vital experiments are essential to make conclusive claims. The authors have generated and used Nia-Pro expression Arabidopsis stable lines in recent publications, for example Casteel et al. Plant Physiology 2015.

Thank you for the suggestions. These experiments are routine but do take time. As requested we generated transgenic lines of Arabidopsis that stably express Nia-Pro:GFP. As you can see (Fig 7a) the data is consistent with the transient expression work. We agree it strengthened our conclusions and now we have a powerful tool to further dissect mechanisms in the lab.

Second concern is the method to determine the subcellular localization of vacuole, which I will explain in detail below.

We added additional experiments as suggested below to further confirm vacuole localization. See our comments below.

Without these experiments the importance is diminished and the appropriateness for Nature Communications will be weakened. In addition, essential controls and information are missing in some experiments (see specifics below).

Specifics

Major:

The authors claimed that Nia-Pro can shuttle from the nucleus to the vacuole when the presence of the aphid vectors during infection. The method they only used is the SNARF-1, Dextran-tagged seminaphthorhodafluor-1, a chemical used to stain vacuole of animal cell. The only supporting data shown in Fig. 4 is of low quality. I am far from being convinced at least. I would strongly suggest the authors reading more papers in area of plant cell biology. Actually, plant researchers would like to use fluorescent protein fused with vacuole protein (FP-probes) to mark organelles. You may use TIP1;1-YFP fusion to mark the tonoplast, spRFP-AFVY fusion to mark the lumen.

We agree that the original images of SNARF-1 staining were not clear. We repeated the experiments and included better images demonstrating colocalization. We also included additional information on the microscopy methods as suggested by reviewer 1. To further demonstrate vacuole localization we performed a second set of microscopy experiments with spRFP-AFVY, as suggested. Images from this experiment were also included in the manuscript (Fig. 2). Finally we also purified vacuoles from control and TuMV/GFP:Nia-Pro infected plants with and without aphid feeding. Proteins were extracted from the vacuoles and incubated with anti-GFP with western blotting (Fig.7f). Nia-Pro:GFP was detected only in the vacuoles in the presence of aphids.

Second, "vacuoles" shown in Fig. 4 are something strange. Normally for a mature tobacco leaf mesophyll cell the central vacuole is occupying up to 90% of a cell's volume.

Meristematic cells contain numerous small provacuoles (resembling animal lysosomes) that arise from fusion of trans-Golgi-derived vesicles (Marty, F. 1999. Plant vacuoles. Plant Cell). The authors need to be very caution to claim Nla-Pro relocalization in vacuoles.

We agree that the original images of SNARF-1 staining were not clear. We repeated the experiments and included better images demonstrating colocalization. See comments above about the additional microscopy and western experiments that were conducted to further confirm vacuole relocalization.

Third, they claimed "The virus must somehow "recognize" the presence of the vector and respond actively". Since they used "recognize", they would like to test with various aphid types effect on this recognize, for example using nonvector aphid or even whitefly as a control.

We performed additional experiments with a non-vector that uses a similar intracellular feeding mode (whiteflies) and with a non-vector that feeds intercellularly and thus more destructively compared to aphids (Leafhopper) (Fig. 6e). We were able to observe some relocalization phenotype for both types of non-vector insects, however relocalization was much lower for non-vector insects and did not correlate with damage, suggesting insect specific relocalization. Good suggestion. Thanks!

Minor:

Materials and methods part

Page 13 add *Nicotiana tabacum* cultivar name or specific accession number. Glurk ?
We use cultivar Glurk. This was added to the methods.

Page 14 line 4 The detailed amino acids for NLS and NES used in the research.
We added the information in the material and methods.

line 9-10. Restrict enzyme sites should be correctly written, SacII
We modified the text as suggested.

Line 14 GFP should in italic.
We modified the text as suggested.

Line 16 pCAMBIA not pCambia.
We modified the text as suggested.

Line 19-20. Why not a specific OD at 600 nm, The agrobacterium density from 0.1-0.3 can result in big difference in protein expression when do agroinfiltration.

We changed the text to report the precise OD.

Reviewers' comments:

Reviewer #1 (Remarks to the Author):

I carefully read the second version of the manuscript submitted by Bak et al., as well as their response to the reviewers who assessed the initial version.

I was one of the reviewers and I found that they have adequately addressed my concerns. In particular, they succeeded in purifying vacuoles and thus have been able to clearly demonstrate that NIa-Pro is found in vacuoles during infection only in the presence of aphids.

I also believe that the authors did a fine job at answering the other comments, particularly so with the experiments involving *A. thaliana* transgenic for NIa-Pro. The fact that NIa-Pro goes to vacuoles during infection and in the presence of aphids also supports the validity of the authors' main conclusion.

Reviewer #3 (Remarks to the Author):

In this revised ms the authors have addressed many of the reviewers' questions and suggestions. The ms is therefore stronger and more convincing.

There are some remaining points that should be addressed.

1. Page 5 vacuole localization: this has been improved with use of other markers; re the time lapse image I am not aware that streaming occurs in vacuoles (is this diffusion?). My concern was that GFP might not fluoresce in the pH of the vacuole and what you were observing was auto-fluorescence. A scan across 400-800nm would give a characteristic spectrum for GFP. This can be done with the SP5 if fitted with spectral detectors and would still be worth doing, but perhaps there is a reference to GFP in vacuole that you can cite instead?
2. Page 6 section GFP:Nia-Pro relocalisation is reversible. Reversible suggests that GFP-Nia-Pro can flip flop between vacuole and cytoplasm, I think this must be re-phrased. The authors do not show that GFP-Nia-Pro comes back out of the vacuole as these cells have been destructively imaged. The results show that the original phenotype is restored on removal of aphids i.e. GFP-Nia-Pro is observed in the nucleus and cytoplasm. I suggest the following 'GFP-NIA-Pro localization to the vacuole occurs only when aphids are given access to plants' or some similar form of words.
3. Fig 3b this figure should give %cells with GFP in vacuole since that is the key organelle in the story and many cells will have GFP outside of the nucleus (as previously discussed by reviewers 1 and 3).
4. Line 172 nuclear 'accumulation' might be a better term since even with NES signal there is some nuclear localization.
5. Line 193-5, remove this sentence as the results were not statistically significant.
6. Page 13 lines 306-310: its good to see this insertion in the text. It seems to me unlikely that non-persistent potyviruses that do not require colonizers for transmission would

manipulate aphid behaviour and more likely its an aphid behaviour (they benefit from increased fecundity).

Reviewer #4 (Remarks to the Author):

In this submitted revised manuscript, the authors have tried to go a step further in trying to find more evidences for the role of NIa-Pro in the tripartite interaction with aphid and plant. They provided more evidences to suggest that the viral protein NIa-pro is relocated from the nucleus to the vacuole upon specific insect feeding. However, the reviewer suggests new important experiments are still need to be performed for solid their claims. There are three major concerns.

1. I still would like to suggest they use transgenic GFP:NIa-Pro lines to do some work instead of transient expression in the whole ms.

2. Fig. 4. For the major claim of relocation of vacuole is essential for increase of aphid fecundity, the authors provided NLS and NES variants of NIa-Pro. Actually, vacuole sorting in plant cell has a sorting system. The author may fuse NIa-Pro vacuole targeting peptide from a Soybean seed storage protein (β -Conglycinin), PLSSILRAFY. Also it is important to check whether NIa-Pro contains vacuole targeting signal. Make vacuole targeting signal deletion mutant. Check the effect on aphid fecundity with these two new variants of NIa-Pro. We would see whether the real vacuole location is

3. It is very interesting that the authors provided evidence to show GFP:NIa-Pro relocalization in the presence of two distinct phloem-feeding insects leafhopper and the whitefly in *N. benthamiana*. From the data they provided, the conclusion they made with insect specific is not well supported. Due to small insect size, *N. benthamiana* is not a good host for whitefly since the plant contains long trichome to prevent tiny whitefly efficient probe or damage. Other treatments such as wounding or small smooth rub, even phytohormones (JA, SA) treatment can make GFP:NIa-Pro relocalization. Anyway, new data will present a potential mechanistic basic for explaining this phenomena of relocalization instead of a "feeling" of insect at current shape. By the way, the reviewer don't think leafhoppers can make probing on *N. benthamiana*. I think it is only tiny bio-force induced damage on *N. benthamiana* like that of whitefly made.

Some minor editing on manuscripts:

1. Pg. 3 Ln70. "Its genome is a single ? 10 kb RNA". What's the character before 10 kb refer to?
2. Pg. 19 Ln440. Same as the above. 20x objective? Final figures
3. Reference part. Pg. 21. Ln500. "Virus Research" should be Virus Res.
4. Reference 51. Please provide a direct web site for this data.

Reviewers' comments:

Reviewer #1 (Remarks to the Author):

I carefully read the second version of the manuscript submitted by Bak et al., as well as their response to the reviewers who assessed the initial version.

I was one of the reviewers and I found that they have adequately addressed my concerns. In particular, they succeeded in purifying vacuoles and thus have been able to clearly demonstrate that Nla-Pro is found in vacuoles during infection only in the presence of aphids.

I also believe that the authors did a fine job at answering the other comments, particularly so with the experiments involving *A. thaliana* transgenic for Nla-Pro. The fact that Nla-Pro goes to vacuoles during infection and in the presence of aphids also supports the validity of the authors' main conclusion.

Thank you for your comments. Your suggestion to purify the vacuoles was an excellent idea and the experiment strengthened our conclusions.

Reviewer #3 (Remarks to the Author):

In this revised ms the authors have addressed many of the reviewers' questions and suggestions. The ms is therefore stronger and more convincing.

There are some remaining points that should be addressed.

Page 5 vacuole localization: this has been improved with use of other markers; re the time lapse image I am not aware that streaming occurs in vacuoles (is this diffusion?). My concern was that GFP might not fluoresce in the pH of the vacuole and what you were observing was auto-fluorescence. A scan across 400-800nm would give a characteristic spectrum for GFP. This can be done with the SP5 if fitted with spectral detectors and would still be worth doing, but perhaps there is a reference to GFP in vacuole that you can cite instead?

Thank you for explaining how to perform the scan in more detail. We did not completely understand your original request, but now we do. Yes there are citations that demonstrate GFP does fluoresce in the pH of the vacuole. We included these citations in the manuscript as requested. We also performed a scan across 380-800nm as requested for the GFP:Nla-Pro without or with aphids present to demonstrate that the fluorescence observed is characteristic of the GFP and not autofluorescence. We created a new supplementary figure with this new information (Supplementary Fig. 3). We also changed 'vacuolar streaming' to 'intra-vacuolar trafficking' to clarify this part.

2. Page 6 section GFP:Nia-Pro relocalisation is reversible. Reversible suggests that GFP-Nia-Pro can flip flop between vacuole and cytoplasm, I think this must be re-phrased. The authors do not show that GFP-Nia-Pro comes back out of the vacuole as these cells have been destructively imaged. The results show that the original phenotype is restored on removal of aphids i.e. GFP-Nia-Pro is observed in the nucleus and cytoplasm. I suggest the following 'GFP-NIA-Pro localization to the vacuole occurs only when aphids are given access to plants' or some similar form of words.

We understand your concern and we changed the text as suggested.

3. Fig 3b this figure should give % cells with GFP in vacuole since that is the key organelle in the story and many cells will have GFP outside of the nucleus (as previously discussed by reviewers 1 and 3).

We modified the axis label in all figures to show % of cells with GFP in vacuole as suggested.

4. Line 172 nuclear 'accumulation' might be a better term since even with NES signal there is some nuclear localization.

We changed the wording in the text as suggested.

5. Line 193-5, remove this sentence as the results were not statistically significant.

We removed the sentence as suggested.

6. Page 13 lines 306-310: its good to see this insertion in the text. It seems to me unlikely that non-persistent potyviruses that do not require colonizers for transmission would manipulate aphid behaviour and more likely its an aphid behaviour (they benefit from increased fecundity).

Thank you for this comment.

Reviewer #4 (Remarks to the Author):

In this submitted revised manuscript, the authors have tried to go a step further in trying to find more evidences for the role of Nla-Pro in the tripartite interaction with aphid and plant. They provided more evidences to suggest that the viral protein Nla-pro is relocated from the nucleus to the vacuole upon specific insect feeding. However, the reviewer suggests new important experiments are still need to be performed for solid their claims. There are three major concerns.

1. I still would like to suggest they use transgenic GFP:Nla-Pro lines to do some work instead of transient expression in the whole ms.

As requested in the previous reviews we demonstrated that Nla-Pro also relocalizes in the presence of aphids in transgenic Arabidopsis. We previously demonstrated that TuMV infection and Nla-Pro expression increases aphid fecundity and inhibits aphid-induced defenses in Arabidopsis (as well as N.benthamiana) in Casteel et al, 2014. We do not think it is necessary to repeat all the experiments from this published work. Constructing transgenic Arabidopsis for all of the other expressions plasmids used in the current study and repeating all experiments in this system is not possible in a reasonable time frame and we believe this data would only provide minimal advances in our understanding. Currently the lab is utilizing transgenic Arabidopsis to further tease apart different aspects of this interaction, however this is beyond the scope of the current paper.

2. Fig. 4. For the major claim of relocation of vacuole is essential for increase of aphid fecundity, the authors provided NLS and NES variants of Nla-Pro. Actually, vacuole sorting in plant cell has a sorting system. The author may fuse Nla-Pro vacuole targeting peptide from a Soybean seed storage protein (β -Conglycinin), PLSSILRAFY. Also it is important to check whether Nla-Pro contains vacuole targeting signal. Make vacuole targeting signal deletion mutant. Check the effect on aphid fecundity with these two new variants of Nla-Pro. We would see whether the real vacuole location is

Thank you for these suggestions. We have previously conducted the bioinformatic analyses requested to determine if Nla-Pro has nuclear or vacuolar targeting peptides (SignalIP, Phobius). No conventional nuclear or vacuole targeting signals were found in Nla-Pro.

We are currently investigating the plant targets in the vacuole, vacuole transport pathways involved in relocalization, and the vacuole targeting signal in Nla-Pro mediating localization, however this work is at least a year away from publication and the focus of another paper. The suggested experiments with vacuole sorting tags are a good idea and will fit nicely in our next paper, thank you for the suggestions!

3. It is very interesting that the authors provided evidence to show GFP:Nla-Pro relocalization in the presence of two distinct phloem-feeding insects leafhopper and the whitefly in N. benthamiana. From the

data they provided, the conclusion they made with insect specific is not well supported. Due to small insect size, *N. benthamiana* is not a good host for whitefly since the plant contains long trichome to prevent tiny whitefly efficient probe or damage. Other treatments such as wounding or small smooth rub, even phytohormones (JA, SA) treatment can make GFP:Nla-Pro relocalization. Anyway, new data will present a potential mechanistic basic for explaining this phenomena of relocalization instead of a “feeling” of insect at current shape. By the way, the reviewer don't think leafhoppers can make probing on *N. benthamiana*. I think it is only tiny bio-force induced damage on *N. benthamiana* like that of whitefly made.

It is true that the long trichomes and the acyl-sugars found in the trichomes on N. Benthamiana deters feeding by many insects including aphids, whiteflies and most likely leafhoppers. However, previous studies have verified that whiteflies and leafhoppers are capable of transmitting viruses to N. Benthamiana (ref 36 and 37) that are not mechanically transmissible. This published data suggests these insects are capable of probing in N. Benthamiana. Further in all of our experiments we caged insects to the lower surface of the leaf, where there are ~80% fewer trichomes compared to the top. On the lower surface of the leaf, insects have adequate room to interact with the plant and probe. To verify that all insects used in our study were capable of probing, we performed additional experiments and added this new data to the paper (Supplementary Fig. 5). We caged aphids, whiteflies, and leafhoppers to the lower side of separate N. benthamiana leaves. Twenty-four hours later leaves were collected and stained with acid fuchsin to visualize salivary sheaths. Salivary sheaths were observed for all insects demonstrating they were able to probe (See Supplementary Fig. 5).

Some minor editing on manuscripts:

1. Pg. 3 Ln70. “Its genome is a single ? 10 kb RNA”. What's the character before 10 kb refer to?

We removed the character, this was a mistake.

2. Pg. 19 Ln440. Same as the above. 20x objective? Final figures

We removed the character, this was a mistake.

3. Reference part. Pg. 21. Ln500. “Virus Research” should be Virus Res.

We modified the reference as suggested.

4. Reference 51. Please provide a direct web site for this data.

We added the whole web site as suggested.

Reviewers' comments:

Reviewer #3 (Remarks to the Author):

The submitted additional experimental work in the revised manuscript has addressed many of the reviewers' comments. The remaining points are:

1. The revised text (abstract and pages 4 line 89-90, 6 lines 144, 7 line 161) now suggests that the GFP-NIa re-localization to the nucleus is the most important feature. However, in your CLSM images even when GFP-NIa is seen in the vacuole there is still some nuclear localisation (e.g. fig 1). I think the key point is that GFP-NIa disappears from the vacuole and the ms should be modified to reflect this.

2. Vacuole purification, Fig 5F; please state in the methods what was done to confirm that vacuoles were actually purified/quality of the preps.

3. Methods page 18, line 413, please add a sentence to clarify that leaves were infiltrated (not plants) and the approx. area of infiltration per leaf.

Reviewers' comments:

Reviewer #3 (Remarks to the Author):

The submitted additional experimental work in the revised manuscript has addressed many of the reviewers' comments. The remaining points are:

1. The revised text (abstract and pages 4 line 89-90, 6 lines 144, 7 line 161) now suggests that the GFP-N1a re-localization to the nucleus is the most important feature. However, in your CLSM images even when GFP-N1a is seen in the vacuole there is still some nuclear localisation (e.g. fig 1). I think the key point is that GFP-N1a disappears from the vacuole and the ms should be modified to reflect this.

Thank you for this comment. We agree that this phrasing would more accurately reflect our data. We changed the wording throughout the MS to reflect this.

2. Vacuole purification, Fig 5F; please state in the methods what was done to confirm that vacuoles were actually purified/quality of the preps.

We confirmed vacuole purity/quality using microscopy throughout the purification process. We added a sentence in the methods and an additional figure in the supplementary data so the reader can access our preps. The following text was added:

The purity and quality of the protoplast isolations and of vacuole isolations were accessed using light microscopy (Supplementary Fig. 6).

3. Methods page 18, line 413, please add a sentence to clarify that leaves were infiltrated (not plants) and the approx. area of infiltration per leaf.

We clarified in the method that an entire leaf was infiltrated for each plant. The following text was added:

*Single leaves from three-week-old *N. benthamiana* or *N. tabacum* plants were then agroinfiltrated with the solution. For *N. benthamiana* entire leaves were infiltrated. For *N. tabacum* a circle of ~5 cm diameter was infiltrated for each leaf.*